# *Aspergillus niger* as a Biological Input for Improving Vegetable Seedling Production

**DOI:** 10.3390/microorganisms10040674

**Published:** 2022-03-22

**Authors:** Gustavo de Souza Marques Mundim, Gabriel Mascarenhas Maciel, Gilberto de Oliveira Mendes

**Affiliations:** Instituto de Ciências Agrárias, Universidade Federal de Uberlândia, Rodovia LMG-746, km 1, Bloco 1A-MC, Monte Carmelo 38500-000, MG, Brazil; gustavosmmundim@gmail.com (G.d.S.M.M.); gabrielmaciel@ufu.br (G.M.M.)

**Keywords:** biofertilizer, seedling production, seedling vigor, plant growth promotion, bio-input

## Abstract

This study evaluated the potential of *Aspergillus niger* as an inoculant for growth promotion of vegetable seedlings. Seven vegetable species were evaluated in independent experiments carried out in 2^2^ + 1 factorial schemes, with two doses of conidia (10^2^ and 10^6^ per plant) applied in two inoculation methods (seed treatment and in-furrow granular application), plus an uninoculated control. Experiments were carried out in a greenhouse. Growth parameters evaluated were shoot length, stem diameter, root volume, total root length, shoot and root fresh mass, shoot and root dry mass, and total dry mass. Regardless of the dose and inoculation method, seedlings inoculated with *A. niger* showed higher growth than uninoculated ones for all crops. The highest relative increase promoted by the fungus was observed for aboveground parts, increasing the production of shoot fresh mass of lettuce (61%), kale (40%), scarlet eggplant (101%), watermelon (38%), melon (16%), pepper (92%), and tomato (42%). *Aspergillus niger* inoculation also increased seedling root growth of lettuce, pepper, scarlet eggplant, watermelon, and tomato. This research shows that *A. niger* boosts the growth of all analyzed vegetables, appearing as a promising bio-input for vegetable seedling production.

## 1. Introduction

The production of seedlings is a critical step in vegetable crops since during the germination and initial development plants are overly sensitive to biotic and abiotic stresses [1,2]. Moreover, the use of high-quality, healthy, and vigorous seedlings is crucial to achieve the crop yield potential. Seedlings represent 14.8% of the production costs of tomato in soilless cultivation [3]. Therefore, commercial seedling production in highly specialized nurseries has become a trend [2].

Vegetable crops require large amounts of phytosanitary products, fertilizers, and water [4]. These practices have caused a progressive decrease in diversity and amount of soil microorganisms [2,5]. Rhizosphere microbiome engineering has been proposed as a way to reinstate beneficial plant-microorganism associations, and thus harness microbial functions in agroecosystems [5,6]. Therefore, inoculation of beneficial microorganisms during seedling production can be a strategy to introduce traits that may improve plant tolerance to biotic and abiotic stresses after transplanting [5,6,7,8]. Furthermore, this approach might allow decreasing the use of external inputs as well as be an option for organic production [9,10,11].

Plant beneficial microorganisms (PBM) can promote plant growth by various mechanisms, including phytohormone production, increasing nutrient availability, enhancing tolerance to salinity and drought, and disease suppression [8,11,12,13,14]. On the other hand, plants can select for different microbial functional traits according to environmental constraints, shaping the associated microbiota through signaling molecules such as indoleacetic acid [15], abscisic acid [16], strigolactones [17], and flavonoids [18]. For example, nutrient availability modulated functional traits recruited by plants so that PBMs showing phytohormone production were favored in rich nutrient soils while phosphate solubilizers predominated in poor nutrient soils [19]. Therefore, PBMs showing multiple mechanisms of plant growth promotion, called multifunctional microorganisms, can help plants to cope with different environmental constraints, representing a promising option for the development of inoculants [8,20]. Fungi are particularly attractive in this regard due to their capacity to produce high amounts of long-lived spores [21], allowing long shelf life of inoculants. Some species of the genera *Aspergillus*, *Penicillium*, *Pythium*, and *Trichoderma* show multiple plant-growth promotion mechanisms [22,23,24,25], but inoculants with fungi are still limited compared to bacteria [26].

*Aspergillus niger* v. Tiegh is a multifunctional fungus capable of phosphate solubilization [27,28,29], potassium solubilization [30], and phytohormone production [23,31,32]. Plants fertilized with phosphate solubilized by *A. niger* showed enhanced growth and P uptake [27,29,33]. Moreover, *A. niger* promoted the growth of coffee (*Coffea arabica* L.) and maize (*Zea mays* L.) seedlings [23,34]. Therefore, we hypothesized that *A. niger* could promote the growth of vegetable seedlings, enabling the production of seedlings with enhanced root system and aboveground parts. This research aimed at evaluating the growth of seedlings of seven vegetable species inoculated with *A. niger* at different doses and inoculation methods.

## 2. Materials and Methods

### 2.1. Experimental Site

The study was carried out at the Horticultural Experimental Station of the Universidade Federal de Uberlândia (UFU), located in the city of Monte Carmelo, Minas Gerais state, Brazil (18°42’43.19″ S, 47°29’55.8″ W, 873 m altitude). The experiments were performed in a greenhouse covered with 150 µm clear plastic film between March and June 2019. Average temperature and relative humidity in the period were 21.5 °C and 79.4%, respectively [35].

### 2.2. Experimental Design

The experiments were carried out in 2^2^ + 1 factorial schemes. Treatments consisted in combinations of two inoculation methods (in-furrow granular application and seed treatment) and two doses of conidia (10^2^ and 10^6^ conidia per plant) of *A. niger* FS1. An additional uninoculated treatment was used as control (Table 1). Seven vegetable crops were evaluated in independent experiments: melon (*Cucumis melo* L., Cucurbitaceae), watermelon (*Citrullus lanatus* Thumb. Mansf., Cucurbitaceae), tomato (*Solanum lycopersicum* L., Solanaceae), pepper (*Capsicum annuum* L., Solanaceae), scarlet eggplant (*Solanum gilo* L., Solanaceae), lettuce *(Lactuca sativa* L., Asteraceae), and kale (*Brassica oleracea* L. var. *acephala*, Brassicaceae). Each experiment was set up in a randomized complete block design with eight repetitions, adding up 40 plots. Each plot contained eight plants, adding up 64 plants per treatment.

Plants were grown in 128-cell polystyrene trays (27.7 cm^3^ cell^−1^) filled with a commercial coconut fiber substrate (Technes, São Paulo, SP, Brazil). *Aspergillus niger* inoculation was performed at sowing by in-furrow application of one granule of the formulation or by seed treatment (see Section 2.3). Two seeds were sown in each cell, except for Cucurbitaceae species which had just one seed per cell due to their high germinating power. After emergence, seedlings were thinned to one per cell. Seedlings were irrigated daily, and nutrients were supplied weekly by fertigation with 13.4 mL per cell of a solution containing (total amount added per cell): 5.02 µg N, 2.19 µg P, 5.55 µg K, 0.83 µg Mg, 0.083 µg Zn, 0.025 µg B, 0.0083 µg Fe, 0.083 µg Mn, and 0.92 µg S.

Experiments were evaluated after the seedling production period for each vegetable, i.e., 17, 18, 32, 33, 35, 37, and 44 days for watermelon, melon, tomato, lettuce, pepper, kale, and scarlet eggplant, respectively. The variables measured were shoot height, stem diameter, shoot fresh and dry mass, root dry mass, root volume (obtained by water displacement in a graduated test tube), and total root length, measured with the software RootReader2D (v4.3, Robert W. Holley Center for Agriculture & Health, Ithaca, NY, USA) [36]. For lettuce, it was determined the number of leaves instead of stem diameter since this species does not present a well-developed stem.

### 2.3. Aspergillus niger Inoculum Preparation

The fungus *A. niger* FS1 was obtained previously from soil under native forest in Viçosa, Minas Gerais state, Brazil (20°46′4.2″ S, 42°52′40.9″ W) [28]. The fungus was maintained on Petri dishes containing potato dextrose agar (PDA, Sigma-Aldrich, Saint Louis, MO, USA) at 30 °C in the dark. Fungal conidia were collected from 10-day old cultures using a Tween 80 0.01% (*v*/*v*) solution. The conidial suspension obtained was vacuum filtered through membranes with 0.45 µm pores and the conidia retained on the membranes were dried in a desiccator with silica gel at room temperature (25 °C) for 24 h [34]. The mass of dry conidia contained 4.5 × 10^7^ conidia mg^−1^ as determined by counting in a Neubauer chamber. Dry conidia were used for producing a granular formulation and seed treatment.

The granular formulation was produced by mixing dry conidia to 26.5 g wheat flour, 3.8 g corn starch, 2.25 g granulated sugar, and 15 mL deionized water to form a homogeneous bulk [34]. The amount of conidia added to the mixture was 11.9 and 119 mg to produce concentrations of 10^2^ and 10^6^ conidia per granule, respectively. These amounts were calculated with respect to the average mass of each granule, which was 22.6 mg granule^−1^. The bulk was extruded through a noodle-maker fitted with a template of 2 mm diameter holes. The noodles were cut into 2 mm long granules, and then dried in an oven with forced air circulation at 50 °C for 48 h [34].

The conidial suspension for seed treatment was prepared by mixing the dry conidia with sterile water 2 h before sowing. The conidial suspension was pipetted over the seeds in a volume enough to cover them and gently mixed to allow homogeneous distribution of conidia on the seeds. Due to the differences in seed size among the vegetable species, the amount of conidia used to prepare the suspension was adjusted according to the volume of water necessary to cover the seeds, so that the final concentration was 10² or 10^6^ conidia per plant.

### 2.4. Statistical Analyses

Data were subjected to ANOVA and post hoc comparisons of most interest were performed using the value of the least significant difference between two means at *p* = 0.05 (LSD 5%), calculated from the standard error of the difference between two means (SED). Multivariate analyses based on all growth parameters were carried out to cluster treatments by a hierarchical method and the Tocher optimization method. Dendrograms were constructed based on the dissimilarity between treatments calculated by Euclidean distance using the package Nbclust for R (RStudio v1.2.5001-3, Boston, MA, USA) [37]. The relative importance of measured variables on the dissimilarity between treatments was calculated as proposed by Singh [38]. Validation of clustering was conducted based on the cophenetic correlation coefficient [39] calculated in the software Genes (v2021.146, Federal University of Viçosa, Viçosa, MG, Brazil) [40].

## 3. Results

Significant differences (*p* < 0.05) between inoculated and uninoculated treatments were observed for all vegetable species (Table 2). In inoculated treatments, generally, there was no difference between the inoculation methods and doses of *A. niger* (Figure 1).

*Aspergillus niger* inoculation increased root fresh mass of pepper, scarlet eggplant, watermelon, and tomato (Table 2). Root dry mass of inoculated lettuce, pepper, and scarlet eggplant was higher than the uninoculated control. *Aspergillus niger* inoculation also increased total root length of all species, except for melon and watermelon. Total root length was not measured for kale since the software RootReader2D was not able to read the root system due to the high density of fine roots.

Inoculated seedlings showed higher aboveground growth compared to uninoculated ones. Shoot fresh mass and shoot height of all species were higher in inoculated treatments (Table 2). Likewise, all species showed higher shoot dry mass and total dry mass when inoculated with *A. niger*, except for melon. Stem diameter of all species was higher in inoculated treatments, except for tomato and kale. Inoculation also increased the number of leaves of lettuce.

Cluster analysis distinguished two groups of treatments, one containing the uninoculated control and the other containing the treatments inoculated with *A. niger* (Figure 2). The only exception was kale, for which the inoculated treatments were further divided into two groups, one with the seed treatment at the dose of 10^6^ conidia plant^−1^ (TS06) and another with the other inoculated treatments (TS02, GR02 e GR06) (Figure 2G). Cophenetic correlation coefficients confirmed the accuracy of the clustering, showing values higher than 0.8 [41] for all species: lettuce (0.97), melon (0.94), pepper (0.99), scarlet eggplant (0.99), watermelon (0.98), tomato (0.97), and kale (0.98), all significant by the *t* test (*p* < 0.01). Likewise, Tocher optimization method showed high similarity between inoculated treatments (GR02, GR06, TS02 e TS06) and high dissimilarity of these with the uninoculated control for all vegetable crops (Figure 3).

The relative importance analysis showed that variables associated with the aboveground growth of seedlings were the main responsible for the dissimilarity observed between treatments (Table 3). Aboveground parameters (shoot fresh mass, shoot height, stem diameter, and shoot dry mass) summed up to 70.3, 78.3, 93.4, 72.2, 90.1, 72.1, and 82.3% of the dissimilarity observed between treatments in the experiments with lettuce, tomato, kale, scarlet eggplant, watermelon, melon, and pepper, respectively (Table 3).

## 4. Discussion

*Aspergillus niger* inoculation promoted the growth of seedlings of all vegetable crops tested, increasing the shoot fresh mass of lettuce (61%), kale (40%), scarlet eggplant (101%), watermelon (38%), melon (16%), pepper (92%), and tomato (42%). Inoculation also enhanced root growth of lettuce, pepper, scarlet eggplant, watermelon, and tomato. The capacity of *A. niger* to produce indoleacetic acid (IAA) and gibberellic acid (GA) has been demonstrated [23,31,32]. These phytohormones regulate cell growth and elongation in plants [42] and thus may be related to the greater growth observed in inoculated seedlings. Microbial production of phytohormones has been repeatedly reported as an important mechanism of plant growth promotion [12,43,44,45,46]. Plant growth-promoting bacteria modified the architecture and functioning of tomato roots due to the production of phytohormones and other metabolites [47,48]. Volatile organic compounds (VOCs) have also been proposed as a mechanism of microbial plant growth promotion. Exposure to VOCs produced by *Trichoderma asperellum* increased the number of leaves and roots, plant biomass and chlorophyll content in lettuce [25]. Some VOCs were detected in *A. niger* cultures [49,50,51], however the potential role of these compounds on plant growth was not studied yet.

*Aspergillus niger* was effective at the different doses of conidia (10^2^ and 10^6^ conidia plant^−1^). Conidial germination in *A. niger* is very efficient, with more than 90% of conidia germinating at suitable conditions [52]. This allows efficient colonization even at low numbers of conidia, which might explain the similarity between the doses of conidia evaluated. Likewise, the inoculation methods (in-furrow granular application and seed treatment) were equally efficient to deliver the fungus to the plant root. While the granular formulation contained organic substrates that could be used by the fungus during its establishment [53], in the seed treatment there was nothing other than water. Thus, our data suggest that the fungus derived organic C from the seedling root exudates or from the substrate. Taken together, these fungal traits can make the inoculation step simpler and more economic, enabling the use of water dispersible formulations, such as a wettable powder, at low doses of conidia.

The effect of *A. niger* inoculation was more pronounced on aboveground parts of seedlings. Shoot mass and height of inoculated seedlings were higher than uninoculated ones for all vegetable crops tested. Aboveground growth parameters showed relative importance varying from 70.3 to 82.3% of the dissimilarity observed between treatments. GA promotes leaf expansion and stem growth and elongation [42,54], and could be involved in the enhanced growth of aboveground parts promoted by *A. niger*. Furthermore, since inoculated seedlings grew more and hence faster, inoculation might be used to reduce seedling production time and thus reducing costs. Moreover, inoculation in nursery can be an easy, economic, and efficient way to introduce the fungus into the field [6]. Once established in the seedling rhizosphere, *A. niger* would have a competitive advantage over native microbiota in the field and thus the growth promotion effect could be extended throughout the crop cycle [34].

Roots are highly responsive to fluctuations in IAA levels, which control the growth of primary and secondary roots as well as the development of adventitious roots [46]. IAA production is a trait frequently showed by plant growth-promoting microorganisms [12,19,44,46], including *A. niger* [23,32]. Therefore, enhanced root growth is a common effect of plant growth-promoting microorganisms [34,55,56,57]. Indeed, *A. niger* inoculation increased seedling root growth of lettuce, pepper, scarlet eggplant, watermelon, and tomato. Seedlings with well-developed root system can explore a higher volume of soil and hence be more efficient in reach nutrients and water in the field [2]. It should be mentioned that seedlings were exposed to root air pruning in the tray [58] and, therefore, root growth was probably underestimated. This could be the reason for the lower effects observed in roots compared to aboveground parts.

This research provides evidence of the potential of *A. niger* as a biofertilizer for vegetable crops. *Aspergillus niger* enhanced the growth of seedlings of seven species of vegetables belonging to four different botanical families—Cucurbitaceae, Solanaceae, Asteraceae, and Brassicaceae. These results suggest that *A. niger* has no specificity towards the host plant, in line with other reports showing growth promotion of coffee (*Coffea arabica* L.) [34] and maize (*Zea mays* L.) [23] seedlings by *A. niger*. This represents an advantage for the development of products with this fungus. Moreover, plants inoculated with *A. niger* would benefit from multiple mechanisms of plant growth promotion, such as phytohormone production [23,31,32] and solubilization of phosphate [27,28,29] and potassium [30]. Microorganisms showing these multifunctional traits may be able to help plants to cope with different environmental stresses [6,8,19,20]. Finally, the use of bio-inputs, such as the one described herein, can contribute to reducing costs in seedling production as well as increase crop productivity sustainably.

## Figures and Tables

**Figure 1 microorganisms-10-00674-f001:**
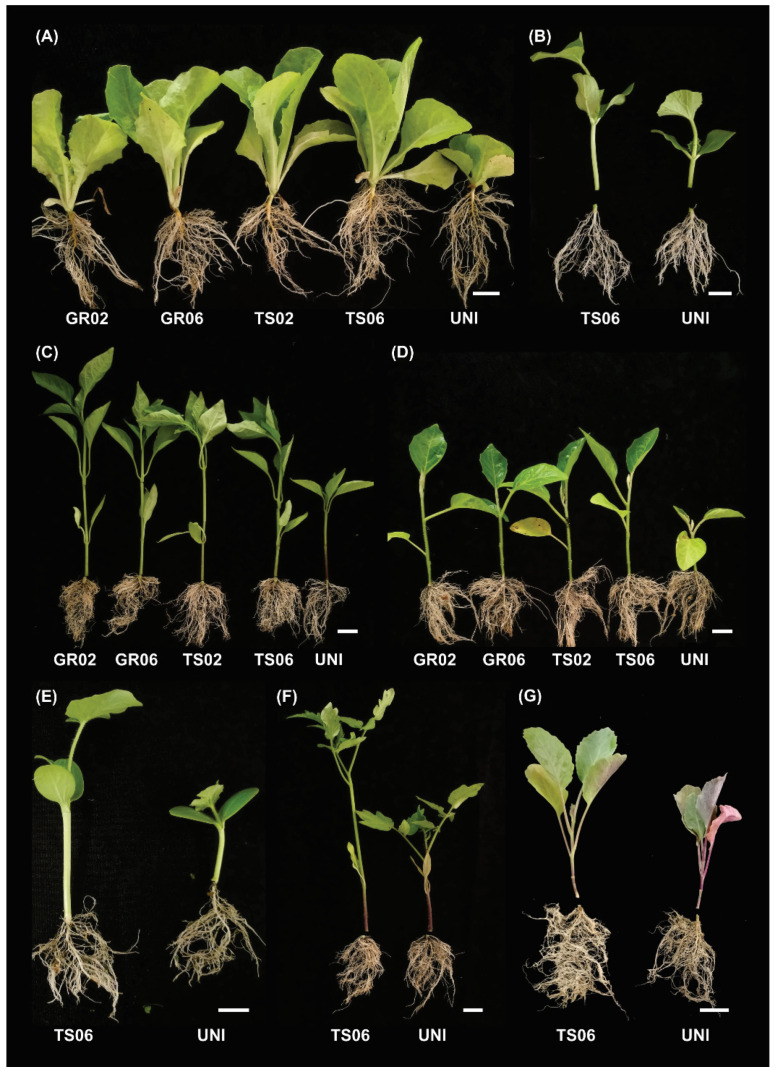
Seedlings of (**A**) lettuce, (**B**) melon, (**C**) pepper, (**D**) scarlet eggplant, **(E)** watermelon, (**F**) tomato, and (**G**) kale inoculated or not with *Aspergillus niger* FS1. GR02 and GR06: in-furrow granular application at 10^2^ and 10^6^ conidia plant^−1^, respectively; TS02 and TS06: seed treatment at 10^2^ and 10^6^ conidia plant^−1^, respectively; UNI: uninoculated control. Scale bars: 2 cm.

**Figure 2 microorganisms-10-00674-f002:**
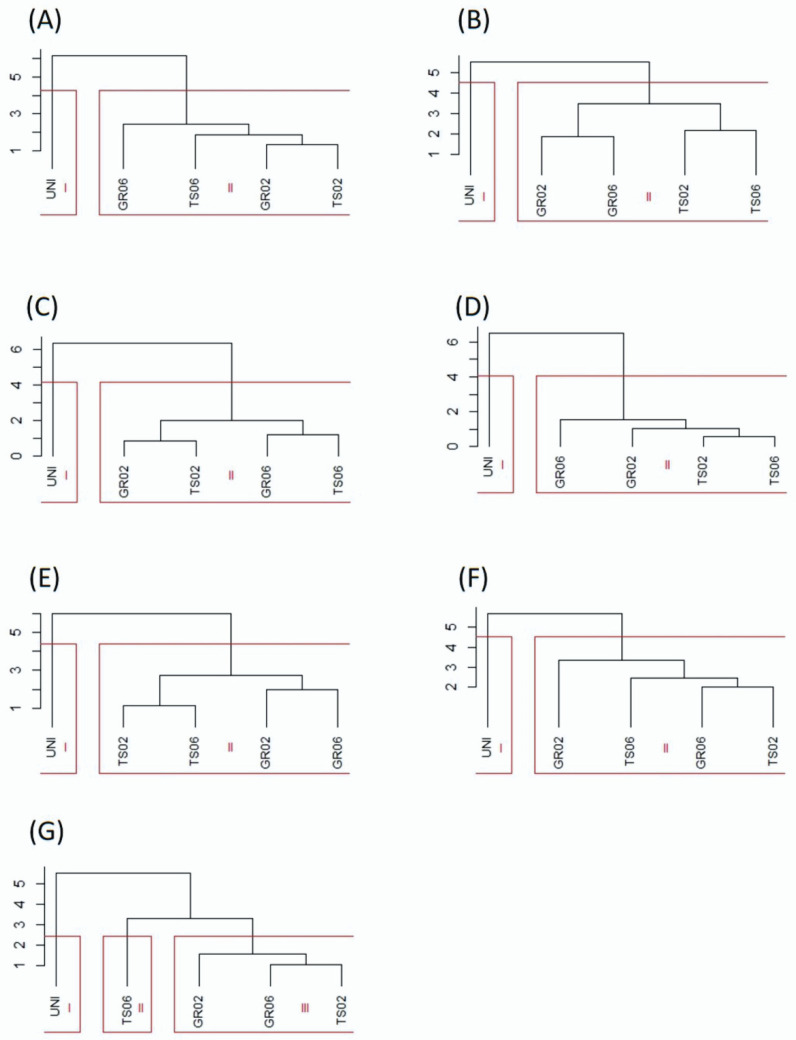
Clustering of treatments with different inoculation methods and doses of *Aspergillus niger* FS1 in vegetables: (**A**) lettuce, (**B**) melon, (**C**) pepper, (**D**) scarlet eggplant, (**E**) watermelon, (**F**) tomato, and (**G**) kale. The vertical axis represents the dissimilarity (%) between groups. GR02 and GR06: in-furrow granular application at 10^2^ and 10^6^ conidia plant^−1^, respectively; TS02 and TS06: seed treatment at 10^2^ and 10^6^ conidia plant^−1^, respectively; UNI: uninoculated control.

**Figure 3 microorganisms-10-00674-f003:**
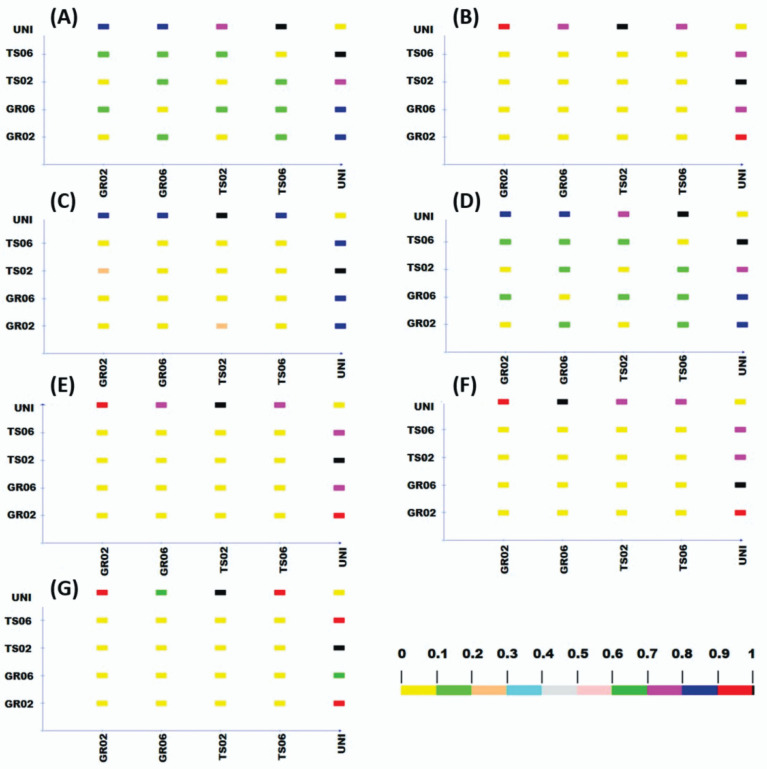
Dissimilarity between treatments with different inoculation methods and doses of *Aspergillus niger* FS1 in vegetables: (**A**) lettuce, (**B**) melon, (**C**) pepper, (**D**) scarlet eggplant, (**E**) watermelon, (**F**) tomato, and (**G**) kale. Color scale determined by Tocher optimization method varies from no dissimilarity (yellow) to complete dissimilarity (black) between each pair of treatments. GR02 and GR06: in-furrow granular application at 10^2^ and 10^6^ conidia plant^−1^, respectively; TS02 and TS06: seed treatment at 10^2^ and 10^6^ conidia plant^−1^, respectively; UNI: uninoculated control.

**Table 1 microorganisms-10-00674-t001:** Combinations of inoculation methods and doses of *Aspergillus niger* in treatments.

Treatment	Inoculation Method	Dose (Conidia Plant^−1^)
GR02	In-furrow granular	4 × 10^2^
GR06	In-furrow granular	4 × 10^6^
TS02	Seed treatment	4 × 10^2^
TS06	Seed treatment	4 × 10^6^
UNI	Uninoculated	0

**Table 2 microorganisms-10-00674-t002:** Effect of inoculation method and dose of *A. niger* FS1 on plant growth parameters of vegetable seedlings.

Treatment	Root Fresh Mass (g)	Shoot Fresh Mass (g)	Shoot Height (cm)	Stem Diameter ^1^ (mm)	Root Volume (cm³)	Total Root Length (cm)	Root Dry Mass (g)	Shoot Dry Mass (g)	Total Dry Mass (g)
Lettuce^1^ (*Lactuca sativa*)									
GR02	0.35	1.13	6.21	5.87	0.35	182	0.037	0.082	0.119
GR06	0.3	1.31	6.58	5.9	0.29	183	0.04	0.096	0.136
TS02	0.33	1.13	6.25	5.91	0.32	182	0.042	0.087	0.129
TS06	0.37	1.2	6.37	6.06	0.36	192	0.039	0.093	0.132
UNI	0.28	0.74	4.82	5.6	0.27	142	0.026	0.058	0.083
SED	0.0462	0.1330	0.4370	0.0993	0.0474	13.9000	0.0043	0.0087	0.0109
LSD 5%	0.0939	0.2703	0.8880	0.2018	0.0963	28.2448	0.0087	0.0177	0.0221
Melon (*Cucumis melo*)									
GR02	0.66	1.58	13.46	3.48	0.51	181	0.028	0.11	0.138
GR06	0.66	1.64	13.03	3.47	0.52	176	0.026	0.106	0.132
TS02	0.69	1.7	13.55	3.49	0.53	186	0.03	0.118	0.148
TS06	0.73	1.65	13.26	3.44	0.53	189	0.028	0.118	0.148
UNI	0.69	1.42	10.49	3.21	0.64	229	0.032	0.112	0.14
SED	0.0759	0.0866	0.5870	0.0839	0.0439	14.5000	0.0023	0.0074	0.0089
LSD 5%	0.1542	0.1760	1.1928	0.1705	0.0892	29.4640	0.0046	0.0149	0.0181
Pepper (*Capsicum annuum*)									
GR02	0.64	1.8900	23.3300	2.8100	0.5100	189.0000	0.0530	0.2000	0.2530
GR06	0.56	1.82	22.9	3.02	0.45	182	0.05	0.195	0.245
TS02	0.62	1.84	22.91	2.7	0.54	192	0.054	0.192	0.246
TS06	0.54	1.73	22.43	2.82	0.45	192	0.049	0.186	0.235
UNI	0.34	0.95	14.91	2.44	0.41	155	0.036	0.109	0.145
SED	0.0426	0.0869	0.7220	0.1200	0.0572	9.5900	0.0038	0.0117	0.0146
LSD 5%	0.0866	0.1766	1.4671	0.2438	0.1162	19.4869	0.0077	0.0238	0.0297
Scarlet eggplant (*Solanum gilo*)									
GR02	0.69	1.46	15.52	2.96	0.67	248	0.07	0.213	0.283
GR06	0.74	1.64	16.37	3.05	0.72	265	0.069	0.233	0.302
TS02	0.71	1.47	15.92	2.94	0.66	262	0.076	0.217	0.293
TS06	0.7	1.46	15.25	3.06	0.66	256	0.075	0.215	0.29
UNI	0.54	0.75	8.01	2.27	0.54	226	0.055	0.112	0.167
SED	0.0526	0.1280	0.6510	0.1230	0.0561	8.2700	0.0039	0.0160	0.0191
LSD 5%	0.1069	0.2601	1.3228	0.2499	0.1140	16.8046	0.0079	0.0325	0.0388
Watermelon (*Citrullus lanatus*)									
GR02	0.79	1.44	12.77	3.31	0.39	168	0.019	0.107	0.126
GR06	0.64	1.47	13.37	3.34	0.4	157	0.018	0.104	0.122
TS02	0.7	1.58	13.8	3.31	0.48	174	0.021	0.11	0.131
TS06	0.74	1.48	13.49	3.26	0.45	172	0.02	0.109	0.129
UNI	0.46	1.08	9.22	3.02	0.37	179	0.023	0.088	0.11
SED	0.0984	0.0688	0.5110	0.0665	0.0446	20.0000	0.0019	0.0041	0.0048
LSD 5%	0.1999	0.1398	1.0384	0.1351	0.0907	40.6400	0.0038	0.0083	0.0097
Tomato (*Solanum lycopersicum*)									
GR02	0.54	2.23	26.58	3.07	0.51	233	0.045	0.254	0.297
GR06	0.55	2.21	26.66	2.98	0.53	205	0.04	0.268	0.308
TS02	0.51	2.33	26.67	3.03	0.48	185	0.041	0.256	0.294
TS06	0.58	2.38	26.55	3.15	0.53	190	0.042	0.262	0.301
UNI	0.4	1.57	18.2	2.97	0.41	195	0.047	0.221	0.261
SED	0.0458	0.1160	0.8070	0.1040	0.0440	8.1900	0.0026	0.0129	0.0145
LSD 5%	0.0931	0.2357	1.6398	0.2113	0.0894	16.6421	0.0053	0.0262	0.0295
Kale (*Brassica oleracea*)									
GR02	0.4	1.7	13.96	2.81	0.42	nd ^2^	0.053	0.25	0.303
GR06	0.42	1.64	13.8	3.02	0.43	nd	0.054	0.259	0.313
TS02	0.42	1.7	14.21	2.7	0.43	nd	0.055	0.259	0.314
TS06	0.36	1.64	14.17	2.82	0.37	nd	0.053	0.259	0.311
UNI	0.41	1.19	11.7	2.44	0.39	nd	0.05	0.22	0.27
SED	0.0431	0.0834	0.2880	0.1270	0.0483	nd	0.0034	0.0118	0.0131
LSD 5%	0.0876	0.1695	0.5852	0.2581	0.0981	nd	0.0068	0.0240	0.0266

^1^ For lettuce, it was measured the number of leaves instead of the stem diameter. ^2^ nd: not determined since the software RootReader2D was not able to read root system. SED: standard error of the difference between two means; LSD 5%: least significant difference between two means at *p* = 0.05 (degrees of freedom = 35). GR02 and GR06: in-furrow granular application at 10^2^ and 10^6^ conidia plant^−1^, respectively; TS02 and TS06: seed treatment at 10^2^ and 10^6^ conidia plant^−1^, respectively; UNI: uninoculated control.

**Table 3 microorganisms-10-00674-t003:** Relative importance (%) of measured variables on the dissimilarity between treatments for each vegetable crop.

Variable	Lettuce	Tomato	Kale	Scarlet Eggplant	Watermelon	Melon	Pepper
Root fresh mass	0	6.19	0.06	0.88	1.58	0.44	4.66
Shoot fresh mass	26.6	9.34	5.71	0.54	19.22	0	10.91
Shoot height	15.97	65.1	5.94	33.99	61.2	57.15	10.83
Stem diameter ^1^	12.9	3.95	0.19	3.56	3.19	14.12	0
Root volume	7.92	0	0.06	1.52	6.84	9.11	0
Total root length	13.75	12.58	nd ^2^	0	1.44	17.62	0.71
Root dry mass	7.98	2.84	6.45	25.33	0	0.66	12.3
Shoot dry mass	0	0	81.59	34.17	0	0.91	60.58
Total dry mass	14.89	0	0	0	6.53	0	0

^1^ For lettuce, it was measured the number of leaves instead of the stem diameter. ^2^ nd: not determined since the software RootReader2D was not able to read root system.

## Data Availability

The dataset generated during the current study is available on the Mendeley Data Repository, DOI: 10.17632/nh6pzmb4tf.1.

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
