# Peer review of "Aspergillus niger as a Biological Input for Improving Vegetable Seedling Production"

_microorganisms, 2022, doi:10.3390/microorganisms10040674_

Round 1

Reviewer 1 Report

I appreciate the interesting topic of the article and the quality of methodology, results, and discussion. I have no significant comments. I can only recommend better emphasizing the results you obtained. In the introduction or discussion, it would be appropriate to emphasize the use of other organisms, such as Pythium oligandrum; further, deal more with the activity of the phytohormones Strigolactones (SLs).

English errors:
Line 28 - Moreover, the use high-quality, healthy, and vigorous seedlings is crucial..

  • The use of?
  • Are crucial

Line 227 - Shoot mass and height of inoculated seedlings was higher

  • Were higher

Line 260 - ... described herein can contribute to reduce costs in seedling production...

  • Contribute to reducing costs – Don’t use a verb in the infinitive after contribute

Author Response

Thank you for your comments and suggestions. Following are our answers:

"I can only recommend better emphasizing the results you obtained".

We emphasized the main results in the Discussion (see lines 209-211).

In the introduction or discussion, it would be appropriate to emphasize the use of other organisms, such as Pythium oligandrum;

Done. See lines 52-56.

further, deal more with the activity of the phytohormones Strigolactones (SLs).

We included a sentence about SL and other plant signaling molecules (see lines 45-46).

English errors:
Line 28 - corrected

Line 227 - corrected

Line 260 - ... corrected

Reviewer 2 Report

This study evaluated the potential of Aspergillus niger as an inoculant for growth promotion in different horticultural crops. Given the good results obtained, it is of great importance for further commercialisation as a bio-stimulant.

Minor Revision

Figure 3 does not look good. it should be sharpened.

lines 147 -152 are not justified

The discussion could be somewhat more extensive. Discussion of possible mechanisms that may favour plant development.

Author Response

Thank you for your comments and suggestions. Following are our answers:

Figure 3 does not look good. it should be sharpened.

We replaced the file with another one with a higher resolution (600 dpi).

lines 147 -152 are not justified

Corrected.

The discussion could be somewhat more extensive. Discussion of possible mechanisms that may favour plant development.

As requested by reviewer 1, we included some evidence from our data to improve the discussion (lines 209-211). Discussion about the possible mechanisms involved is presented in lines 211-223, 239-240, 248-254, and 265-269.